# Effects of pericapsular nerve group block versus local anesthetic infiltration for postoperative analgesia in total hip arthroplasty: A protocol for systematic review and meta-analysis

Lingzhi Rong[1☯], Tangqi Qin[2☯], Shoujia Yu[2], Donghang Zhang[3], Yiyong Wei(iD)[4]*

1 Department of Anesthesiology, Longgang District Central Hospital of Shenzhen, Shenzhen, China,
2 Department of Anesthesiology, The Second Affiliated Hospital, School of Medicine, The Chinese University of Hong Kong, Shenzhen & Longgang District People's Hospital of Shenzhen, Shenzhen, China,
3 Department of Anesthesiology, West China Hospital, Sichuan University, Chengdu, China, 4 Department of Anesthesiology, Longgang Maternity and Child Institute of Shantou University Medical College (Longgang District Maternity & Child Healthcare Hospital of Shenzhen City), Shenzhen, China

☯ These authors contributed equally to this work.
* 295502476@qq.com

## Abstract

### Introduction

This protocol for a systematic review and meta-analysis aims to provide synthesized evidence to determine whether pericapsular nerve group (PENG) block is superior to local anesthetic infiltration in controlling postoperative pain in total hip arthroplasty.

### Methods and analysis

PubMed, EMBASE, Web of science, and the Cochrane library will be systematically searched from their inception to December 30, 2024. Randomized controlled trials (RCTs) that compared the analgesic effects of PENG block with local anesthetic infiltration for total hip arthroplasty will be included. The time to first analgesics requirement (analgesia duration) will be the primary outcome. Secondary outcomes will include the postoperative analgesics consumption over 24 hours, visual analog scale (VAS) scores at rest and movement, and the incidence of adverse effects. Statistical analysis will be conducted by RevMan 5.4 software.

### Ethics and dissemination

Ethical approval is not applicable. The results of this study will be publicly published.

### PROSPERO registration number

CRD42024590888

**Data availability statement:** No datasets were generated or analysed during the current study. All relevant data from this study will be made available upon study completion.

**Funding:** This work was supported by the Guangdong Basic and Applied Basic Research Foundation (grant No. 2024A1515012880, YW). This funder contributed to the study conceptualization, validation, manuscript writing and editing.

**Competing interests:** The authors have declared that no competing interests exist.

## Introduction

Total hip arthroplasty is one of the most common types of orthopedic surgeries [1]. Management of postoperative pain is one major challenge after total hip arthroplasty [2]. Local anesthetic infiltration is generally performed near the completion of total hip arthroplasty by injecting long-acting local anesthetics in the surgical wound and surrounding tissues, which is suggested to provide effective pain relief and reduce the opioids consumption [3–5]. Recently, pericapsular nerve group (PENG) block is emerging as a novel technique targeting the sensory branch of the anterior hip capsule, and is gaining popularity to control postoperative pain after total hip arthroplasty [6–8]. Increasing number of RCTs demonstrated that PENG block improves postoperative pain, decrease the analgesics requirement or prolong the time to first analgesia requirement [9–11]. More importantly, PENG block can provide motor-sparing analgesia, which facilitated the functional recovery after total hip arthroplasty [12]. Additionally, the PENG block can be achieved in the supine position, which is particularly convenient for patients with acute hip fractures [13]. Several systematic review and meta-analysis have also indicated that, compared with the placebo or control group, PENG block was effective in controlling postoperative pain, extended analgesia duration, and reduced opioid consumption after total hip arthroplasty [6,14]. Therefore, PENG block is recommended as one important component of multimodal analgesia [12,15]. In 2015, a network meta-analysis has made an indirect comparison of the analgesic effects between PENG block and local anesthetic infiltration for total hip arthroplasty [16]. Recently, several studies have directly compared the analgesic effects of PENG block and local anesthetic infiltration for total hip arthroplasty [10,17–20], but the results are conflicting. To the best of our knowledge, no systematic review and meta-analysis that directly compare the analgesic effects of PENG block with local anesthetic infiltration has yet been published. Therefore, it is worthwhile to perform a systematic review and meta-analysis to determine whether PENG block is superior to local anesthetic infiltration in postoperative analgesia for total hip arthroplasty.

## Methods and analysis

### Study registration

We have registered this protocol in the International Prospective Register of Systematic Reviews (CRD42024590888). This study was constructed in line with the Preferred Reporting Items for Systematic Evaluation and Meta-Analysis Protocols (PRISMA-P) guidelines. The PRISMA-P-checklist is described in S1 File. Ethical approval is not applicable.

### Search strategy

Two independent authors will systematically search four databases including PubMed, EMBASE, Web of science, and the Cochrane library with the following key terms: "pericapsular nerve group block", "local anesthetic infiltration", "local infiltration analgesia", "total hip arthroplasty", and "randomized controlled trials". The search time will be set from their inception to December 30, 2024. The language will be restricted to English. The search plan for all databases was presented in S2 File.

### Inclusion and exclusion criteria

Inclusion criteria: 1) Study type: RCTs; 2) Participants: patients underwent total hip arthroplasty; 3) Interventions: pericapsular nerve group block; 4) Control: local anesthetic infiltration; 5) Primary outcomes: the time to first analgesics requirement (analgesia duration); Secondary outcomes: postoperative analgesics consumption over 24 hours, visual analog scale (VAS) scores at

rest and movement, and the incidence of adverse effects. Studies do not meet above-mentioned criteria will be excluded, which include the following study types: retrospective studies, systematic review and meta-analysis, narrative reviews, conference abstracts, case reports, comments, letters, perspectives, insights, correspondences, and editorials. Moreover, in order to further optimize the quality of included studies, studies with an extremely small sample size or extremely low quality or unavailable full-text, and repeatedly published studies will be excluded.

## Study selection

Firstly, two independent authors will read the titles and abstracts of initially identified studies. Then, the full text of potentially relevant studies will be reviewed for inclusion. When we encountered a study with incomplete information during the study selection process, we will contact the authors for additional information to determine whether the study meet the inclusion criteria. Disagreements will be solved by discussion with a third author. The detailed process for study selection was presented in Fig 1.

## Data extraction

Two independent authors will perform data extraction from the included studies, including published date, regions, characteristics of patients, sample number, anesthesia type, local anesthetics, adjuvants, comparisons, and outcomes. Any discrepancy will be solved by discussion with a third author.

## Risk of bias assessment

Two independent authors will perform the risk of bias assessment for included studies using the Cochrane Collaboration's tool. The risk of bias will be rated as 'unclear', 'low' or 'high' according to the estimated results of six items, including random sequence generation (selection bias), allocation concealment (selection bias), blinding of participants and personnel (performance bias), blinding of outcome assessment (detection bias), incomplete outcome data (attrition bias), and selective reporting (reporting bias). Disagreements will be solved by discussion with a third author.

## Statistical analysis

RevMan 5.4 will be used to perform the statistical analysis. Mean differences (MD) and risk ratios (RR) with 95% confidence intervals (CI) will be used for continuous and dichotomous variables, respectively. Statistical heterogeneity will be calculated by $I^2$ test. The fixed-effect model will be applied when $I^2 < 50\%$. The random-effect model will be used when $I^2 > 50\%$. $P < 0.05$ represents statistical significance.

## Subgroup analysis

If $I^2 > 50\%$, subgroup analysis will further be performed to explore the source of heterogeneity based on several potential factors, such as types of participants, types of local anesthetics, the definition of primary outcomes, the combination drugs, language restrictions, etc. A meta-analysis will be performed and the summary effects will be computed within subgroups. According to the results of subgroups, we can recommend a preferred analgesia method for specific conditions.

## Sensitivity analysis

Sensitivity analysis will be used to assess the reliability of the pooled results by excluding or including studies based on sample size, methodological quality, or variance. Sensitivity analysis

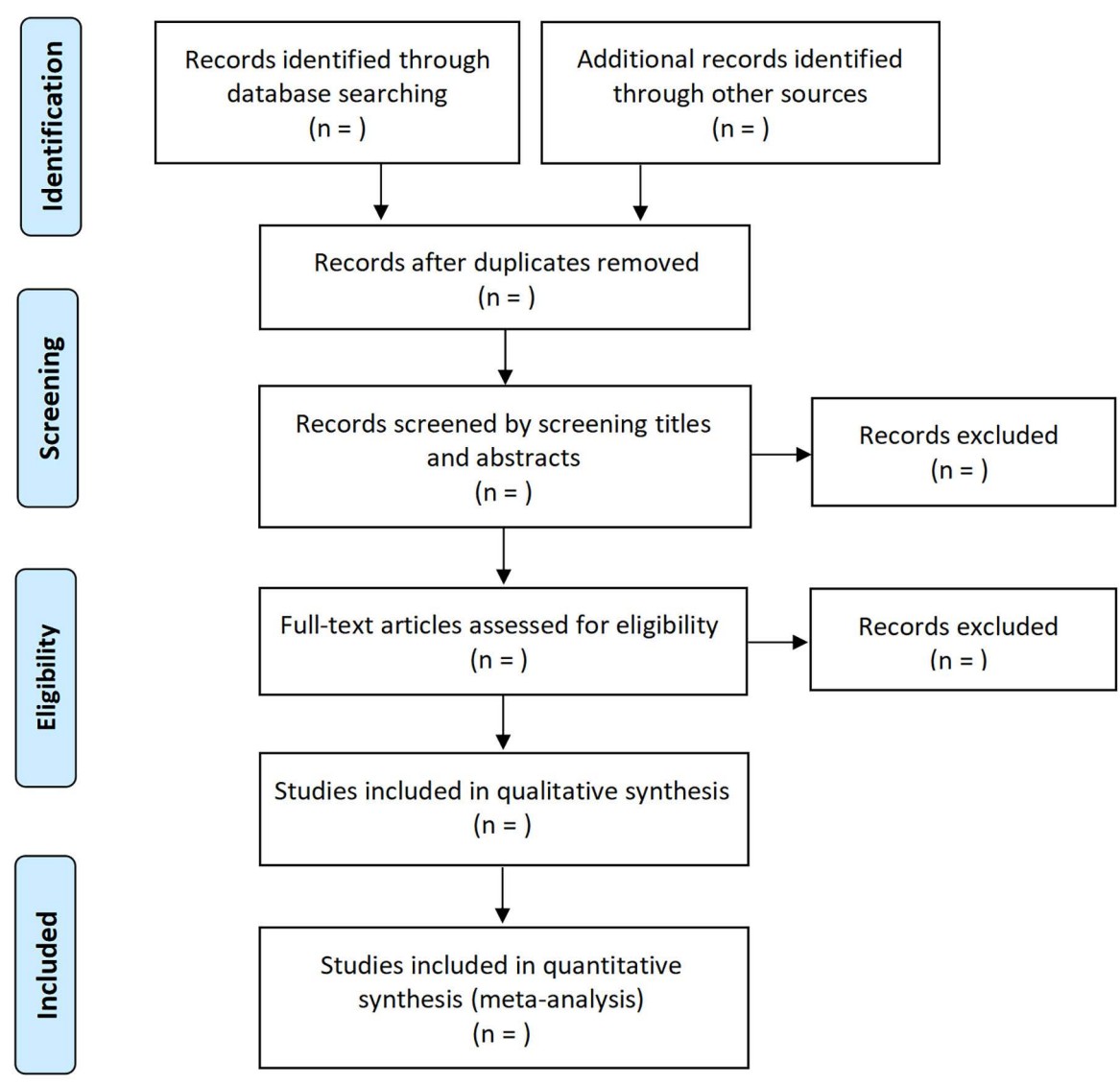

**Fig 1. The flowchart of study selection.**

may also explore the impact of using different meta-analysis models. If the pooled results remain consistent across different analyses, the results can be considered reliable. Conversely, the results should be interpreted with caution when an inconsistence exists across sensitivity analyses.

## Publication bias

Egger's test will be used to assess potential publication bias via the funnel plots, which are a scatterplot of each study's effect size on the x-axis plotted against its standard error on the y-axis. A symmetrical upside-down funnel with smaller studies at the top and more studies at the bottom indicates no publication bias. A skewed funnel indicates publication bias exists, and a 'trim and fill' method will be further used to correct the funnel plot asymmetry by 1) removing the smaller studies causing asymmetry, 2) using the trimmed funnel plot to evaluate the true funnel center, and 3) replacing the omitted studies around the funnel center.

## Evidence quality assessment

The Grading of Recommendations Assessment, Development and Evaluation (GRADE) approach will be used to assess the evidence quality of pooled results and will create a 'Summary of findings' table. The GRADE approach will classify the quality of evidence into four levels:

High: there is a lot of confidence that the true effect lies close to that of the estimated effect.

Moderate: there is moderate confidence in the estimated effect, which means the true effect is likely to be close to the estimated effect, but there is a possibility that it is substantially different.

Low: there is limited confidence in the estimated effect, which means the true effect might be substantially different from the estimated effect.

Very low: there is very little confidence in the estimated effect, which means the true effect is likely to be substantially different from the estimated effect.

Detailed information could be found in Cochrane handbook (https://training.cochrane.org).

## Patient and public involvement

Patients and/or the public were not involved in the design, or conduct, or reporting, or dissemination plans of this research.

## Discussion

Although increasing numbers of studies have compared the effects of PENG block with local anesthetic infiltration on postoperative pain for total hip arthroplasty, no meta-analysis has yet provided synthesized evidence. This study aimed to provide a protocol to determine whether PENG block is superior to local anesthetic infiltration in controlling postoperative pain for total hip arthroplasty. There might be several limitations when conducting this meta-analysis. First, substantial heterogeneity between included studies resulting from the type of local anesthetics, the definition of primary outcomes, the combination drugs, regional differences regarding medical levels and patients' characteristics, differences in follow-up time, etc., will influence the reliability of pooled results. Second, the number of RCTs that directly compared the effects of PENG block to local anesthetic infiltration might be small. Finally, publication bias might exist because the English language restriction. To minimize the influence of heterogeneity, subgroup analysis will be used to explore their source, and sensitivity analysis will be performed to assess the reliability of pooled data. For subsequent studies, it's better to include patients with similar baselines, including medical levels, characteristics, and follow-up time, etc. Furthermore, the GRADE approach will be conducted to rank the evidence quality for major outcomes. This protocol has been registered in the PROSPERO and was generated according to the PRISMA-P guidelines.

Exploratory Data Analysis might be performed when encountering unexpected patterns or variations in outcomes. For example, different types of analgesics might be used across included studies, and we will unify the analgesic consumption to morphine equivalent consumption according to the methods described in previous papers (e.g., morphine 1 mg, iv. = tramadol 10 mg, iv. = fentanyl 10 mcg, iv. = sufentanil 1 mcg, iv = pethidine 10 mg, iv = oxycodone 1.5 mg, oral) [21,22]. For data that presented using the median and range, we will convert them to the mean and standard deviation [23]. If there were two interventions groups in one included study, we will combine them into one intervention group (Cochrane

Handbook for Systematic Reviews of Interventions Version 5.1.0.). If pain scores were not reported clearly at rest or on movement, we will contact the authors.

## Strengths and limitations of this study

- This protocol has been registered in the PROSPERO and was generated according to the PRISMA-P guidelines.

- Two authors will independently perform the databases search, study selection, data extraction, and risk of bias assessment.

- Substantial heterogeneity might exist between included studies resulting from the type of local anesthetics, the definition of primary outcomes, the combination drugs, etc.

- Publication bias might exist because the English language restriction.

- Subgroup analysis and sensitivity analysis will be used to explore the source of heterogeneity, and the GRADE approach will be conducted to rank the evidence quality for major outcomes.

## Supporting information

**S1 File. The PRISMA-P-checklist.**
(DOC)

**S2 File. Search strategy for all databases.**
(DOCX)

## Author contributions

**Conceptualization:** Donghang Zhang, Yiyong Wei.

**Methodology:** Lingzhi Rong, Tangqi Qin, Shoujia Yu, Donghang Zhang.

**Validation:** Yiyong Wei.

**Writing – original draft:** Lingzhi Rong, Tangqi Qin, Shoujia Yu, Donghang Zhang, Yiyong Wei.

**Writing – review & editing:** Yiyong Wei.

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
