## [Decision Letter · Decision Letter 0]

4 Nov 2024

PONE-D-24-45094Effects of pericapsular nerve group block versus local anesthetic infiltration for postoperative analgesia in total hip arthroplasty: a protocol for systematic review and meta-analysisPLOS ONE

Dear Dr. Wei,

Thank you for submitting your manuscript to PLOS ONE. After careful consideration, we feel that it has merit but does not fully meet PLOS ONE’s publication criteria as it currently stands. Therefore, we invite you to submit a revised version of the manuscript that addresses the points raised during the review process. Please revise. Please submit your revised manuscript by Dec 19 2024 11:59PM. If you will need more time than this to complete your revisions, please reply to this message or contact the journal office at plosone@plos.org . Please include the following items when submitting your revised manuscript:

We look forward to receiving your revised manuscript.

Kind regards,

Academic Editor

PLOS ONE

Journal Requirements: When submitting your revision, we need you to address these additional requirements. 1. Please ensure that your manuscript meets PLOS ONE's style requirements, including those for file naming. The PLOS ONE style templates can be found at https://journals.plos.org/plosone/s/file?id=wjVg/PLOSOne_formatting_sample_main_body.pdf and https://journals.plos.org/plosone/s/file?id=ba62/PLOSOne_formatting_sample_title_authors_affiliations.pdf 2. Thank you for stating the following financial disclosure: "This work was supported by the Guangdong Basic and Applied Basic Research Foundation (grant No. 2024A1515012880, YW)." Please state what role the funders took in the study.  If the funders had no role, please state: ""The funders had no role in study design, data collection and analysis, decision to publish, or preparation of the manuscript."" If this statement is not correct you must amend it as needed. Please include this amended Role of Funder statement in your cover letter; we will change the online submission form on your behalf.

Reviewers' comments:

Reviewer's Responses to Questions

**Comments to the Author**

1. Does the manuscript provide a valid rationale for the proposed study, with clearly identified and justified research questions?

Reviewer #1: Yes

Reviewer #2: Partly

2. Is the protocol technically sound and planned in a manner that will lead to a meaningful outcome and allow testing the stated hypotheses?

Reviewer #1: Partly

Reviewer #2: Partly

3. Is the methodology feasible and described in sufficient detail to allow the work to be replicable?

Reviewer #1: Yes

Reviewer #2: No

4. Have the authors described where all data underlying the findings will be made available when the study is complete?

Reviewer #1: No

Reviewer #2: Yes

5. Is the manuscript presented in an intelligible fashion and written in standard English?

Reviewer #1: Yes

Reviewer #2: Yes

6. Review Comments to the Author

You may also provide optional suggestions and comments to authors that they might find helpful in planning their study.

Reviewer #1: 1) The manuscript outlines a rationale by highlighting the conflicting results in existing studies regarding the efficacy of pericapsular nerve group (PENG) block versus local anesthetic infiltration for postoperative pain control in total hip arthroplasty. The study aims to perform a systematic review and meta-analysis to synthesize available evidence, addressing a significant gap in the literature and contributing to the optimization of pain management strategies in orthopedic surgery. A similar meta analysis comparing placebo Vs LIA vs PENG was done previously in 2016.

(Jiménez-Almonte, et. al, Is Local Infiltration Analgesia Superior to Peripheral Nerve Blockade for Pain Management After THA: A Network Meta-analysis. Clinical Orthopaedics and Related Research 474(2):p 495-516, February 2016. | DOI: 10.1007/s11999-015-4619-9)

2) The protocol outlines appropriate databases and search strategies, including clear inclusion and exclusion criteria. The study design aligns with PRISMA-P guidelines, ensuring methodological rigor. However, potential limitations, such as language restrictions and variations in analgesic types and outcome definitions, may impact the reliability of results. These aspects are acknowledged but could benefit from a more comprehensive strategy for mitigating bias.

3) The methodology includes detailed criteria for study selection, data extraction, and statistical analysis. The protocol specifies the use of established tools (e.g., RevMan 5.4) for meta-analysis, risk assessment methods, and criteria for handling heterogeneity. The detailed approach enhances reproducibility.

4) The manuscript states that no datasets were generated during the protocol’s development and that relevant data will be available upon study completion. However, it does not specify the exact repository or access details for data sharing, which is required for transparency and compliance with data availability policies.

5) The protocol is clearly written, with standard scientific English and minimal typographical errors. The structure follows accepted guidelines for study protocols, enhancing readability and comprehension.

6) The manuscript provides a solid foundation for a systematic review and meta-analysis aimed at resolving discrepancies in postoperative analgesia strategies for total hip arthroplasty.

However, there are a few areas for improvement:

a) Data Availability: Specify the repository and data access policies to ensure transparency.

b) Mitigation of Bias: Address limitations related to language restrictions and heterogeneity more comprehensively, potentially by planning subgroup analyses or sensitivity checks.

c) Exploratory Aspects: Consider discussing how exploratory analyses will be handled, particularly if unexpected patterns or variations in outcomes are found.

Overall, the study protocol appears fair, with potential to contribute good insights into pain management techniques for orthopedic surgery.

Reviewer #2: 1. This research has a certain degree of innovation and has certain reference value for clinical practice. However, the manuscript is very brief and there are few references. It is recommended to further enrich the content. For example, subgroup analysis, sensitivity analysis, publication bias analysis, and the method of GRADE assessment of quality grades should be introduced in detail in separate paragraphs.

2. Please describe in detail what you did when you encountered a study with incomplete information during the study selection process, such as whether you tried to determine whether it met the inclusion criteria by obtaining more information, such as contacting the authors.

3. Please provide the retrieval formula construction process for each database, not just an example of the PubMed retrieval plan, to demonstrate the comprehensiveness and scientific nature of the retrieval.

7. PLOS authors have the option to publish the peer review history of their article (what does this mean? ). If published, this will include your full peer review and any attached files.

**Do you want your identity to be public for this peer review?** For information about this choice, including consent withdrawal, please see our Privacy Policy .

Reviewer #1: **Yes: ** Syed Azfar

Reviewer #2: No

---

## [Author Response · Author response to Decision Letter 1]

8 Nov 2024

Responses to Reviewers/Editor Letter for revision of PONE-D-24-45094

Dear Dr. Robert Jeenchen Chen and Reviewers (Dr. Syed Azfar and reviewer #2),

Thank you very much for giving us the opportunity to revise our manuscript (PONE-D-24-45094). We appreciate the helpful feedback from you and the reviewers. After carefully reading the comments, we have revised the manuscript point-by-point. Herewith we resubmit a revised manuscript for your assessment. Important changes are highlighted, and detailed responses to each comment are included below. We believe that these revisions have substantially improved the manuscript, and we thank you and the reviewers for their thoughtful comments, which are really helpful for the improvement. All authors of the manuscript have read the revised manuscript prior to re-submission and agree with its contents.

We greatly appreciate your consideration of our manuscript and look forward to hearing from you and reviewers.

Journal Requirements:

Response: Thank you! We have checked our manuscript and ensure it meets PLOS ONE's style requirements.

"This work was supported by the Guangdong Basic and Applied Basic Research Foundation (grant No. 2024A1515012880, YW)."

Response: Thank you! We have stated what role the funders took in the study (page 11, line 12-14).

Reviewers' comments:

Reviewer #1: 1) The manuscript outlines a rationale by highlighting the conflicting results in existing studies regarding the efficacy of pericapsular nerve group (PENG) block versus local anesthetic infiltration for postoperative pain control in total hip arthroplasty. The study aims to perform a systematic review and meta-analysis to synthesize available evidence, addressing a significant gap in the literature and contributing to the optimization of pain management strategies in orthopedic surgery. A similar meta analysis comparing placebo Vs LIA vs PENG was done previously in 2016.

(Jiménez-Almonte, et. al, Is Local Infiltration Analgesia Superior to Peripheral Nerve Blockade for Pain Management After THA: A Network Meta-analysis. Clinical Orthopaedics and Related Research 474(2):p 495-516, February 2016. | DOI: 10.1007/s11999-015-4619-9)

Response: Thank you very much for this encouraging comments. In this revision, we have described and cited this study (PMID: 26573322) in this revision (page 4, line 20-22).

2) The protocol outlines appropriate databases and search strategies, including clear inclusion and exclusion criteria. The study design aligns with PRISMA-P guidelines, ensuring methodological rigor. However, potential limitations, such as language restrictions and variations in analgesic types and outcome definitions, may impact the reliability of results. These aspects are acknowledged but could benefit from a more comprehensive strategy for mitigating bias.

Response: We thank the reviewer for this good comment. We completely agree with the reviewer that these potential limitations, including language restrictions and variations in analgesic types and outcome definitions will impact the reliability of combined results. Subgroup analysis will further be performed to explore the source of heterogeneity, and sensitivity analysis will be performed to test the reliability of combined results. We have described this and provided more information for mitigating bias (page 8, line 4-18).

3) The methodology includes detailed criteria for study selection, data extraction, and statistical analysis. The protocol specifies the use of established tools (e.g., RevMan 5.4) for meta-analysis, risk assessment methods, and criteria for handling heterogeneity. The detailed approach enhances reproducibility.

Response: Thank you very much for this encouraging comments.

4) The manuscript states that no datasets were generated during the protocol’s development and that relevant data will be available upon study completion. However, it does not specify the exact repository or access details for data sharing, which is required for transparency and compliance with data availability policies.

Response: Thank you for this nice comment. For this protocol, no datasets have been generated and/or analyzed. When the related meta-analysis of this protocol is finished, we will submit the results and data to one peer-reviewed journal and published it publicly. We have made a Data Availability Statement in this revision (page 11, line 19-24).

5) The protocol is clearly written, with standard scientific English and minimal typographical errors. The structure follows accepted guidelines for study protocols, enhancing readability and comprehension.

Response: Thank you very much.

6) The manuscript provides a solid foundation for a systematic review and meta-analysis aimed at resolving discrepancies in postoperative analgesia strategies for total hip arthroplasty.

Response: Thank you very much for this nice comment.

However, there are a few areas for improvement:

a) Data Availability: Specify the repository and data access policies to ensure transparency.

Response: Thank you. We have specified the repository and data access policies to ensure transparency (page 11, line 19-24).

b) Mitigation of Bias: Address limitations related to language restrictions and heterogeneity more comprehensively, potentially by planning subgroup analyses or sensitivity checks.

Response: Thank you very much for this helpful suggestion. We have described how to address limitations related to language restrictions and heterogeneity more comprehensively by planning subgroup analyses and sensitivity analysis (page 8, line 4-18).

c) Exploratory Aspects: Consider discussing how exploratory analyses will be handled, particularly if unexpected patterns or variations in outcomes are found.

Response: Thank you very much. We have discussed how exploratory analyses will be handled as “Exploratory Data Analysis might be performed when encountering unexpected patterns or variations in outcomes. For example, different types of analgesics might be used across included studies, and we will unify the analgesic consumption to morphine equivalent consumption according to the methods described in one previous paper (PMID: 18574361). For data that presented using the median and range, we will convert them to the mean and standard deviation (PMID: 15840177). If there were two interventions groups in one included study, we will combine them into one intervention group (Cochrane Handbook for Systematic Reviews of Interventions Version 5.1.0.). If pain scores were not reported clearly at rest or on movement, we will contact the authors.” (page 10, line 18-26).

Overall, the study protocol appears fair, with potential to contribute good insights into pain management techniques for orthopedic surgery.

Response: Thank you very much for your valuable comments, which have substantially improved our manuscript.

Reviewer #2: 1. This research has a certain degree of innovation and has certain reference value for clinical practice. However, the manuscript is very brief and there are few references. It is recommended to further enrich the content. For example, subgroup analysis, sensitivity analysis, publication bias analysis, and the method of GRADE assessment of quality grades should be introduced in detail in separate paragraphs.

Response: Thank you very much. We have added detailed information for subgroup analysis, sensitivity analysis, publication bias analysis, and the GRADE methods (page 8, line 4-28; page 9, line 1-14). Moreover, we have described how to handle unexpected patterns or variations in outcomes (page 10, line 18-26) and added more references as recommended (page 4, line 7, 9, 20 and 22).

2. Please describe in detail what you did when you encountered a study with incomplete information during the study selection process, such as whether you tried to determine whether it met the inclusion criteria by obtaining more information, such as contacting the authors.

Response: Thank you very much for this good question. When we encountered a study with incomplete information during the study selection process, we will contact the authors for additional information to determine whether the study meet the inclusion criteria. We have described this (page 7, line 3-5). Furthermore, we have described how to handle unexpected patterns or variations in outcomes (page 10, line 18-26)

3. Please provide the retrieval formula construction process for each database, not just an example of the PubMed retrieval plan, to demonstrate the comprehensiveness and scientific nature of the retrieval.

Response: Thank you for this helpful suggestion. We have included the full search strategy for all databases in a supplementary file (S2_File). Accordingly, this supplementary file has be cited in the main text (page 6, line 15).

---

## [Decision Letter · Decision Letter 1]

20 Nov 2024

PONE-D-24-45094R1Effects of pericapsular nerve group block versus local anesthetic infiltration for postoperative analgesia in total hip arthroplasty: a protocol for systematic review and meta-analysisPLOS ONE

Dear Dr. Wei,

Thank you for submitting your manuscript to PLOS ONE. After careful consideration, we feel that it has merit but does not fully meet PLOS ONE’s publication criteria as it currently stands. Therefore, we invite you to submit a revised version of the manuscript that addresses the points raised during the review process.

Please revise.

We look forward to receiving your revised manuscript.

Kind regards,

Academic Editor

PLOS ONE

Journal Requirements:

Reviewers' comments:

Reviewer's Responses to Questions

**Comments to the Author**

1. Does the manuscript provide a valid rationale for the proposed study, with clearly identified and justified research questions?

Reviewer #1: Yes

Reviewer #2: Yes

2. Is the protocol technically sound and planned in a manner that will lead to a meaningful outcome and allow testing the stated hypotheses?

Reviewer #1: Yes

Reviewer #2: Yes

3. Is the methodology feasible and described in sufficient detail to allow the work to be replicable?

Reviewer #1: Yes

Reviewer #2: Yes

4. Have the authors described where all data underlying the findings will be made available when the study is complete?

Reviewer #1: Yes

Reviewer #2: Yes

5. Is the manuscript presented in an intelligible fashion and written in standard English?

Reviewer #1: Yes

Reviewer #2: Yes

6. Review Comments to the Author

You may also provide optional suggestions and comments to authors that they might find helpful in planning their study.

Reviewer #1: Please accept revised version of submission. As most of the points are well taken and revised as per suggestions.

Reviewer #2: Appreciate the efforts of the authors. The manuscript has been well revised, but there are still some issues that need to be addressed.

1. The description of the exclusion criteria could be more detailed. For example, in addition to the studies that do not meet the inclusion criteria, whether certain specific types of studies are also excluded (such as those with an extremely small sample size or extremely low quality), and the specific reasons for the exclusion. Consider whether it is necessary to set some additional exclusion criteria, such as studies that have been repeatedly published, studies for which the full text cannot be obtained, etc., in order to further optimize the quality of the included studies.

2. If publication bias is found to exist, what measures will be taken to adjust or explain it? For example, conducting a trim-and-fill analysis or exploring the extent to which publication bias affects the research results, so as to enhance the credibility of the research results.

3. Discussion Section.

In addition to the limitations already mentioned, other factors that may affect the research results can also be considered, such as regional differences in the research (the medical level and patient characteristics may vary in different regions), differences in follow-up time (the assessment of long-term analgesic effects and complications may be insufficient), etc., and discuss how to make improvements in subsequent studies.

You mentioned in the text that the usage amount of analgesics will be uniformly converted into the morphine equivalent usage amount. Please elaborate on how you will carry out this conversion.

7. PLOS authors have the option to publish the peer review history of their article (what does this mean? ). If published, this will include your full peer review and any attached files.

**Do you want your identity to be public for this peer review?** For information about this choice, including consent withdrawal, please see our Privacy Policy .

Reviewer #1: No

Reviewer #2: No

---

## [Author Response · Author response to Decision Letter 2]

23 Nov 2024

Responses to Reviewers/Editor Letter for revision of PONE-D-24-45094R1

Dear Dr. Robert Jeenchen Chen and Reviewers,

Thank you very much for giving us the opportunity to revise our manuscript (PONE-D-24-45094R1). We appreciate the helpful feedback from you and the reviewers. After carefully reading the comments, we have revised the manuscript point-by-point. Herewith we resubmit a revised manuscript for your assessment. Important changes are highlighted, and detailed responses to each comment are included below. We believe that these revisions have substantially improved the manuscript, and we thank you and the reviewers for their thoughtful comments, which are really helpful for the improvement. All authors of the manuscript have read the revised manuscript prior to re-submission and agree with its contents.

We greatly appreciate your consideration of our manuscript and look forward to hearing from you and reviewers.

Journal Requirements:

Response: Thank you! We have checked the reference list to ensure that it is complete and correct.

Reviewers' comments:

Reviewer #1: Please accept revised version of submission. As most of the points are well taken and revised as per suggestions.

Response: Thank you very much for your valuable comments, which have substantially improved our manuscript.

Reviewer #2: Appreciate the efforts of the authors. The manuscript has been well revised, but there are still some issues that need to be addressed.

1. The description of the exclusion criteria could be more detailed. For example, in addition to the studies that do not meet the inclusion criteria, whether certain specific types of studies are also excluded (such as those with an extremely small sample size or extremely low quality), and the specific reasons for the exclusion. Consider whether it is necessary to set some additional exclusion criteria, such as studies that have been repeatedly published, studies for which the full text cannot be obtained, etc., in order to further optimize the quality of the included studies.

Response: We thank the reviewer for these nice suggestions. We have added this information in the manuscript as “Studies do not meet above-mentioned criteria will be included, which include the following study types: retrospective studies, systematic review and meta-analysis, narrative reviews, conference abstracts, case reports, comments, letters, perspectives, insights, correspondences, and editorials. Moreover, in order to further optimize the quality of included studies, studies with an extremely small sample size or extremely low quality or unavailable full-text, and repeatedly published studies will be excluded.” (page 6, line 26-28; page 7, line 1-3).

2. If publication bias is found to exist, what measures will be taken to adjust or explain it? For example, conducting a trim-and-fill analysis or exploring the extent to which publication bias affects the research results, so as to enhance the credibility of the research results.

Response: Thank you very much for this good question. We have described how to handle publication bias as “A skewed funnel indicates publication bias exists, and a ‘trim and fill’ method will be further used to correct the funnel plot asymmetry by 1) removing the smaller studies causing asymmetry, 2) using the trimmed funnel plot to evaluate the true funnel center, and 3) replacing the omitted studies around the funnel center.” (page 9, line 1-5).

3. Discussion Section.

In addition to the limitations already mentioned, other factors that may affect the research results can also be considered, such as regional differences in the research (the medical level and patient characteristics may vary in different regions), differences in follow-up time (the assessment of long-term analgesic effects and complications may be insufficient), etc., and discuss how to make improvements in subsequent studies.

Response: Thank you very much for pointing out this issue. We completely agree with the reviewer that other factors that may affect the research results, such as regional differences and differences in follow-up time, we have added this information. Subgroup group analysis and sensitivity analysis will be further conducted to identify their influence on the pooled results. For subsequent studies, it’s better to include patients with similar medical levels, characteristics, and follow-up time. We have discussed this issue in the revision (page 10, line 9-10, line 13-17).

You mentioned in the text that the usage amount of analgesics will be uniformly converted into the morphine equivalent usage amount. Please elaborate on how you will carry out this conversion.

Response: Thank you very much. We have provided the conversion formula for commonly used analgesics and cited the relevant references (page 10, line 24-26).

Finally, thank you again for your great help in improving our manuscript.

---

## [Decision Letter · Decision Letter 2]

9 Dec 2024

PONE-D-24-45094R2Effects of pericapsular nerve group block versus local anesthetic infiltration for postoperative analgesia in total hip arthroplasty: a protocol for systematic review and meta-analysisPLOS ONE

Dear Dr. Wei,

Thank you for submitting your manuscript to PLOS ONE. After careful consideration, we feel that it has merit but does not fully meet PLOS ONE’s publication criteria as it currently stands. Therefore, we invite you to submit a revised version of the manuscript that addresses the points raised during the review process.

Please revise.

We look forward to receiving your revised manuscript.

Kind regards,

Robert Jeenchen Chen, MD, MPH, ChFC®, EA, CLU

Academic Editor

PLOS ONE

Journal Requirements:

Reviewers' comments:

Reviewer's Responses to Questions

**Comments to the Author**

1. Does the manuscript provide a valid rationale for the proposed study, with clearly identified and justified research questions?

Reviewer #3: Yes

Reviewer #4: Partly

2. Is the protocol technically sound and planned in a manner that will lead to a meaningful outcome and allow testing the stated hypotheses?

Reviewer #3: Yes

Reviewer #4: Yes

3. Is the methodology feasible and described in sufficient detail to allow the work to be replicable?

Reviewer #3: Yes

Reviewer #4: Yes

4. Have the authors described where all data underlying the findings will be made available when the study is complete?

Reviewer #3: Yes

Reviewer #4: Yes

5. Is the manuscript presented in an intelligible fashion and written in standard English?

Reviewer #3: No

Reviewer #4: Yes

6. Review Comments to the Author

You may also provide optional suggestions and comments to authors that they might find helpful in planning their study.

Reviewer #3: Thank you very much for allowing me to review your manuscript. I appreciate your effort in performing this study and submitting it to plos one. I would like to point out a few things that should be corrected:

1) I assume the author should rewrite the inclusion part in a better format. In addition, you add a sentence that should change it. You wrote: “Studies do not meet above-mentioned criteria will be included”. I think you wanted to write excluded instead of included.

2) The PRISMA chart you included is a template and I cannot see the number of studies you included or excluded in your PRISMA chart.

3) Your discussion part is too short. Please extend it.

Reviewer #4: The efforts of the authors is praiseworthy.

Kindly shorten the introduction section of abstract. It is lengthy.

7. PLOS authors have the option to publish the peer review history of their article (what does this mean? ). If published, this will include your full peer review and any attached files.

**Do you want your identity to be public for this peer review?** For information about this choice, including consent withdrawal, please see our Privacy Policy .

Reviewer #3: No

Reviewer #4: No

---

## [Author Response · Author response to Decision Letter 3]

11 Dec 2024

Responses to Reviewers/Editor Letter for revision of PONE-D-24-45094R2

Dear Dr. Robert Jeenchen Chen and Reviewers,

Thank you very much for giving us the opportunity to revise our manuscript (PONE-D-24-45094R2). We appreciate the helpful feedback from you and the reviewers. After carefully reading the comments, we have revised the manuscript point-by-point. Herewith we resubmit a revised manuscript for your assessment. Important changes are highlighted, and detailed responses to each comment are included below. We believe that these revisions have substantially improved the manuscript, and we thank you and the reviewers for their thoughtful comments, which are really helpful for the improvement. All authors of the manuscript have read the revised manuscript prior to re-submission and agree with its contents.

We greatly appreciate your consideration of our manuscript and look forward to hearing from you and reviewers.

Journal Requirements:

Response: Thank you! We have checked the reference list to ensure that it is complete and correct.

Reviewers' comments:

Reviewer #3: Thank you very much for allowing me to review your manuscript. I appreciate your effort in performing this study and submitting it to plos one. I would like to point out a few things that should be corrected:

1) I assume the author should rewrite the inclusion part in a better format. In addition, you add a sentence that should change it. You wrote: “Studies do not meet above-mentioned criteria will be included”. I think you wanted to write excluded instead of included.

Response: Thank you very much for your valuable comments. During last revision stage, we have updated the inclusion and exclusion criteria as “Inclusion criteria: 1) Study type: RCTs; 2) Participants: patients underwent total hip arthroplasty; 3) Interventions: pericapsular nerve group block; 4) Control: local anesthetic infiltration; 5) Primary outcomes: the time to first analgesics requirement (analgesia duration); Secondary outcomes: postoperative analgesics consumption over 24 hours, visual analog scale (VAS) scores at rest and movement, and the incidence of adverse effects. Studies do not meet above-mentioned criteria will be excluded, which include the following study types: retrospective studies, systematic review and meta-analysis, narrative reviews, conference abstracts, case reports, comments, letters, perspectives, insights, correspondences, and editorials. Moreover, in order to further optimize the quality of included studies, studies with an extremely small sample size or extremely low quality or unavailable full-text, and repeatedly published studies will be excluded.”. (page 6, line 17-28)

2) The PRISMA chart you included is a template and I cannot see the number of studies you included or excluded in your PRISMA chart.

Response: Thank you very much for this nice question. This meta-analysis has not yet started, therefore, we did not include the number of studies in the PRISMA chart. It is normal for a protocol and is consistent with the journal guidelines of Plos One. Thanks for your understanding.

3) Your discussion part is too short. Please extend it.

Response: Thank you very much. During the revision, we have extended the discussion part as “Although increasing numbers of studies have compared the effects of PENG block with local anesthetic infiltration on postoperative pain for total hip arthroplasty, no meta-analysis has yet provided synthesized evidence. This study aimed to provide a protocol to determine whether PENG block is superior to local anesthetic infiltration in controlling postoperative pain for total hip arthroplasty. There might be several limitations when conducting this meta-analysis. First, substantial heterogeneity between included studies resulting from the type of local anesthetics, the definition of primary outcomes, the combination drugs, regional differences regarding medical levels and patients’ characteristics, differences in follow-up time, etc., will influence the reliability of pooled results. Second, the number of RCTs that directly compared the effects of PENG block to local anesthetic infiltration might be small. Finally, publication bias might exist because the English language restriction. To minimize the influence of heterogeneity, subgroup analysis will be used to explore their source, and sensitivity analysis will be performed to assess the reliability of pooled data. For subsequent studies, it’s better to include patients with similar baselines, including medical levels, characteristics, and follow-up time, etc. Furthermore, the GRADE approach will be conducted to rank the evidence quality for major outcomes. This protocol has been registered in the PROSPERO and was generated according to the PRISMA-P guidelines.

Exploratory Data Analysis might be performed when encountering unexpected patterns or variations in outcomes. For example, different types of analgesics might be used across included studies, and we will unify the analgesic consumption to morphine equivalent consumption according to the methods described in previous papers (e.g., morphine 1 mg, iv. = tramadol 10 mg, iv. = fentanyl 10 mcg, iv. = sufentanil 1 mcg, iv = pethidine 10 mg, iv = oxycodone 1.5 mg, oral). For data that presented using the median and range, we will convert them to the mean and standard deviation. If there were two interventions groups in one included study, we will combine them into one intervention group (Cochrane Handbook for Systematic Reviews of Interventions Version 5.1.0.). If pain scores were not reported clearly at rest or on movement, we will contact the authors.”. (page 10-11)

Reviewer #4: The efforts of the authors is praiseworthy.

Response: Thank you very much for your encouraging comment.

Kindly shorten the introduction section of abstract. It is lengthy.

Response: Thank you very much for this helpful suggestion. We have shortened the introduction section of abstract as “This protocol for a systematic review and meta-analysis aims to provide synthesized evidence to determine whether pericapsular nerve group (PENG) block is superior to local anesthetic infiltration in controlling postoperative pain in total hip arthroplasty.”. (page 2, line 2-5)

Finally, thank you again for your great help in improving our manuscript.

---

## [Decision Letter · Decision Letter 3]

28 Jan 2025

Effects of pericapsular nerve group block versus local anesthetic infiltration for postoperative analgesia in total hip arthroplasty: a protocol for systematic review and meta-analysis

PONE-D-24-45094R3

Dear Dr. Wei,

We’re pleased to inform you that your manuscript has been judged scientifically suitable for publication and will be formally accepted for publication once it meets all outstanding technical requirements.

Kind regards,

Robert Jeenchen Chen, MD, MPH, ChFC®, EA, CLU

Academic Editor

PLOS ONE

Additional Editor Comments (optional):

Reviewers' comments:

Reviewer's Responses to Questions

**Comments to the Author**

1. Does the manuscript provide a valid rationale for the proposed study, with clearly identified and justified research questions?

Reviewer #4: Yes

Reviewer #5: Yes

2. Is the protocol technically sound and planned in a manner that will lead to a meaningful outcome and allow testing the stated hypotheses?

Reviewer #4: Yes

Reviewer #5: Yes

3. Is the methodology feasible and described in sufficient detail to allow the work to be replicable?

Reviewer #4: Yes

Reviewer #5: Yes

4. Have the authors described where all data underlying the findings will be made available when the study is complete?

Reviewer #4: No

Reviewer #5: Yes

5. Is the manuscript presented in an intelligible fashion and written in standard English?

Reviewer #4: Yes

Reviewer #5: Yes

6. Review Comments to the Author

You may also provide optional suggestions and comments to authors that they might find helpful in planning their study.

Reviewer #4: Congratulations to the authors for your hard and dedicated work. The paper is supposed to add value to existing literature.

Reviewer #5: I sincerely appreciate the opportunity to review this article. It is not very common to read a protocol for a systematic review or meta-analysis, as protocols are more often written for original research. However, from a methodological point, this protocol is well-structured and appropriate for publication in Plos One, given its clear design and clinical relevance.

The protocol addresses an important comparison in pain management for total hip arthroplasty, a topic of interest in regional anesthesia. The proposed methodology is adequate, with a clear definition of inclusion and exclusion criteria, as well as a well-designed statistical analysis plan.

In my opinion, the article effectively synthesizes the necessary methodological information for this type of study and meets the quality standards required for publication in Plos One.

7. PLOS authors have the option to publish the peer review history of their article (what does this mean? ). If published, this will include your full peer review and any attached files.

**Do you want your identity to be public for this peer review?** For information about this choice, including consent withdrawal, please see our Privacy Policy .

Reviewer #4: **Yes: ** Dr Satish Prasad Barnawal

Reviewer #5: No

---

## [Editor Report · Acceptance letter]

PONE-D-24-45094R3

PLOS ONE

Dear Dr. Wei,

I'm pleased to inform you that your manuscript has been deemed suitable for publication in PLOS ONE. Congratulations! Your manuscript is now being handed over to our production team.

Kind regards,

on behalf of

Dr. Robert Jeenchen Chen

Academic Editor

PLOS ONE